# Roles Played by Biomarkers of Kidney Injury in Patients with Upper Urinary Tract Obstruction

**DOI:** 10.3390/ijms21155490

**Published:** 2020-07-31

**Authors:** Satoshi Washino, Keiko Hosohata, Tomoaki Miyagawa

**Affiliations:** 1Department of Urology, Jichi Medical University Saitama Medical Center, 1-847, Amanuma-cho, Omiya-ku, Saitama 330-8503, Japan; sh2-miya@jichi.ac.jp; 2Education and Research Center for Clinical Pharmacy, Osaka University of Pharmaceutical Sciences, 4-20-1 Nasahara, Takatsuki 569-1094, Japan; hosohata@gly.oups.ac.jp

**Keywords:** upper urinary tract obstruction, kidney injury, biomarkers, neutrophil gelatinase-associated lipocalin, monocyte chemotactic protein-1, kidney injury molecule 1, cystatin C, vanin-1

## Abstract

Partial or complete obstruction of the urinary tract is a common and challenging urological condition caused by a variety of conditions, including ureteral calculi, ureteral pelvic junction obstruction, ureteral stricture, and malignant ureteral obstruction. The condition, which may develop in patients of any age, induces tubular and interstitial injury followed by inflammatory cell infiltration and interstitial fibrosis, eventually impairing renal function. The serum creatinine level is commonly used to evaluate global renal function but is not sensitive to early changes in the glomerular filtration rate and unilateral renal damage. Biomarkers of acute kidney injury are useful for the early detection and monitoring of kidney injury induced by upper urinary tract obstruction. These markers include levels of neutrophil gelatinase-associated lipocalin (NGAL), monocyte chemotactic protein-1, kidney injury molecule 1, N-acetyl-b-D-glucosaminidase, and vanin-1 in the urine and serum NGAL and cystatin C concentrations. This review summarizes the pathophysiology of kidney injury caused by upper urinary tract obstruction, the roles played by emerging biomarkers of obstructive nephropathy, the mechanisms involved, and the clinical utility and limitations of the biomarkers.

## 1. Introduction

Upper urinary tract obstruction (UUTO) is a common and challenging urological condition caused by a variety of diseases, such as ureteropelvic junction obstruction (UPJO), ureteral calculi, ureteral strictures, and malignant ureteral obstruction. The condition may occur in patients of any age. Surgical intervention is necessary for moderate to severe cases, depending on the cause of the obstruction.

Hydronephrosis, or swelling in one or both kidneys due to incomplete emptying, is often observed in UUTO patients. However, the extent of hydronephrosis does not necessarily reflect the severity of UUTO. Obstruction may be minimal despite moderate to severe hydronephrosis, or it may be severe without obvious hydronephrosis. Renal scans together with determination of the glomerular filtration rate constitute the standard method of evaluating the presence and severity of UUTO. These examinations can be time-consuming and distressing especially to the child, and are not sensitive or specific enough to identify those kidneys that require treatment in all cases. Additionally, renal scans are expensive and not always available. Therefore, there is a great need for the development of new methods to stratify and monitor patients, and the biomarker research field is a promising approach for this purpose. Urinary as well as serum proteins provide information of the physiological condition in the kidney and have the potential to be used as prognostic tools for early disease detection and the choice of the optimal treatment and monitoring [1]. The present review summarizes the pathophysiology of kidney injury caused by UUTO, the roles played by emerging biomarkers of obstructive nephropathy, the mechanisms involved, and the clinical utility and limitations of the biomarkers.

## 2. Upper Urinary Tract Obstruction

### 2.1. UPJO

Congenital obstructive nephropathy reflects maldevelopment of the urinary tract in utero. Most commonly, lesions lie in the ureteropelvic junction (UPJ), causing chronic renal failure. Rapid diagnosis and treatment are essential to preserving function and slowing renal damage. The prevalence is one in 1500 live births [2]. Although UPJO is less common in adults, the condition is not rare [3]. In addition to having a congenital cause, acquired stenosis of the UPJ may follow an upper urinary tract infection, the development of stones, trauma, or ischemia. Vessels that compress or distort the UPJ when crossing the urinary tract may obstruct ureteral outflow in adults.

### 2.2. Ureteral Calculi

Urinary tract stones are very common. The prevalence is 1–19.1% in Asia, 5–9% in Europe, and 7–13% in North America [4,5]. Although most small stones pass spontaneously, some do not, causing UUTO with or without infection. Surgical intervention (shock wave or ureteroscopic lithotripsy) is required to prevent the impairment of renal function [6]. Even after stones are removed, some patients develop ureteral strictures that may continue to impair renal function.

### 2.3. Ureteral Strictures

A ureteral stricture is a narrowing of the ureter that causes an obstruction. Strictures cause significant morbidity and mortality from renal failure. Benign strictures are typically caused by ischemia or inflammation. Causes include radiation, trauma associated with calculus impaction, pelvic surgery, and ureteroscopy [7]. Moderate to severe strictures require surgical intervention, such as balloon dilation, endoureterotomy, or stricture resection.

### 2.4. Malignant Ureteral Obstruction

A malignant ureteral obstruction develops secondary to a malignant tumor. A primary tumor may infiltrate the ureteral wall and compress the ureter, swollen lymph nodes may wrap around the ureter, edema and retroperitoneal fibrosis that develop after radiotherapy may distort the ureter or cause luminal stenosis, or ureter elasticity may be weakened [8]. The condition may be unilateral or bilateral. Clinical removal of the obstruction and rapid improvement in renal function are the aims of treatment. Although ureteral stenting or nephrostomy is performed in severe cases, these procedures reduce the quality of life. Markers of severity are required.

## 3. The Pathophysiology of Kidney Injury Caused by UUTO (Figure 1)

Urinary tract obstruction affects renal function in many ways. The increase in intratubular hydrostatic pressure [9] triggers renopathogenic effects via three proposed mechanisms: tubular ischemia caused by hypoperfusion, pressure-induced mechanical stretching or compression of tubular cells, and altered urinary shear stress. The latter two mechanisms are likely the primary causes of obstructive renal injury [10], being associated with the dysregulation of many cytokines, growth factors, enzymes, and cytoskeletal proteins. Changes in early renal hemodynamics are followed by structural and functional changes in the entire nephron. The earliest stage of UUTO is associated with an increase in renal blood flow 1–2 h in duration [10]. The intrarenal renin–angiotensin–aldosterone system is then activated, which causes pre- and post-glomerular vasoconstriction and resultant drops in renal blood flow, medullary oxygen tension, and the glomerular filtration rate [11,12]. The increased intra-renal angiotensin II activates nuclear factor kappa B, triggering cytokine release and reactive oxygen species (ROS) production [2,10,13,14]. Adhesion molecules such as selectins attract infiltrating macrophages, monocyte chemotactic protein-1 (MCP-1) is upregulated, and tumor necrosis factor-α (TNF-α) is released. Monocytes and macrophages are attracted to the tubular interstitium of the UUTO kidney [2,14,15]. Activated macrophages infiltrate the interstitium, sustaining the inflammatory response by releasing cytokines such as transforming growth factor-β1 (TGF-β1) and TNF-α and ROS [16]. ROS mediate the profibrotic action of TGF-β1, and renal fibrosis proceeds via the epithelial–mesenchymal transition (EMT) of renal tubular epithelial cells. The outcome is interstitial fibrosis caused by increased deposition of the extracellular matrix, cellular infiltration, tubular apoptosis, and the EMT [17]. The mechanical stretching of tubular cells, ischemia, and oxidative stress that follow ureteral obstruction cause tubular cell death [18,19]. Mild injury triggers apoptosis, while tubulointerstitial atrophy after obstruction causes cell deletion. The apoptotic bodies are phagocytosed by neighboring tubular cells or directly shed into the tubular lumen, reestablishing homeostasis. When the injury is severe, necrosis is likely to be the predominant cause of cell loss [19,20]. Increased apoptosis and/or necrosis activates cell infiltration, interstitial cell proliferation, and interstitial fibrosis (Figure 1).

Hydrostatic pressure stretches tubular cells and creates urinary shear stress, inducing intrarenal angiotensin II activation followed by the release of cytokines and adhesion molecules, in turn triggering macrophage infiltration, the production of reactive oxygen species (ROS), and decreased renal blood flow (RBF). The drop in RBF triggers renal ischemia, and the increase of hydrostatic pressure and ROS causes tubular cell death. Monocyte chemotactic protein-1 (MCP-1), tumor necrosis factor-α (TNF-α) and transforming growth factor-β1 (TGF-β1) released from activated macrophages, ROS, and/or tubular cell death induce the epithelial–mesenchymal transition (EMT) and fibroblast proliferation. Eventually the renal parenchyma is transformed into fibrotic tissue.

## 4. Imaging Studies and Their Limitations

Technetium 99m (^99m^Tc) mertiatide, ^99m^Tc diethylene triamine penta-acetic acid, or ^99m^Tc dimercaptosuccinic acid renal scans (with or without diuresis) are commonly used to evaluate the presence and severity of UUTO in patients with hydronephrosis. Patients are divided into those with no, partial, or complete obstruction and with or without renal function [21,22,23]. Urgent surgical relief of complete obstruction is essential, otherwise the kidney will rapidly become nonfunctional. A partial obstruction is a resistance to outflow that, if left untreated, will lead to a loss of kidney function. If renal function is lost, surgery is not considered unless the kidney may be infected. Although renal scans are the standard method of evaluating the presence and severity of UUTO, they are expensive and expose patients to radiation, and repeat scans should be avoided. Furthermore, they do not reveal kidney damage per se, and the equipment is not widely available.

## 5. Biomarkers of UUTO

Urinary and serum biomarkers facilitate the evaluation of renal damage in UUTO patients. An ideal biomarker is assessed noninvasively in a simple manner, is highly sensitive and specific in terms of early detection, and exhibits a wide dynamic range and cutoff values, allowing for risk stratification. Diagnostic utility improves when pelvic urine samples (compared to bladder urine) are used [23,24]. However, the collection of renal pelvic urine is invasive, requiring the placement of an indwelling ureteral catheter via cystoscopy under X-ray guidance, and thus it is difficult to repeatedly collect renal pelvic urine. Biomarkers of kidney injury evaluate glomerular function and renal tubular damage. Serum creatinine (SCr) and cystatin C are representative glomerular function biomarkers. Levels of neutrophil gelatinase-associated lipocalin (NGAL), MCP-1, kidney injury molecule 1 (KIM-1), N-acetyl-b-D-glucosaminidase (NAG), and liver type fatty acid-binding protein (L-FABP) are used to evaluate proximal tubule damage.

### 5.1. Biomarkers of Glomerular Function

#### 5.1.1. SCr

SCr concentrations are widely used to assess kidney function. However, accumulating evidence indicates that measurements of SCr levels do not always detect kidney disease early, and individual variability in SCr generation rates limits the utility of these tests in terms of identifying and assessing the severity of kidney injury [25]. Furthermore, UUTO often affects the unilateral upper urinary tract. The contralateral kidney can compensate for the loss of renal function.

#### 5.1.2. Cystatin C

Cystatin C is an endogenous cysteine protease inhibitor of molecular weight 13.3 kDa secreted by most nucleated cells [26]. It is an ideal filtration marker, being produced at a stable rate, freely filtered without tubular secretion, and completely catabolized in the proximal tubule [27]. Cystatin C is distributed only in the extracellular space and thus reflects changes in the glomerular filtration rate more precisely than creatinine, which is distributed in all body water [28]. In one study, the serum cystatin C level strongly predicted all-cause acute kidney injury (AKI). The area under the curve (AUC; the receiver operating characteristic curve [ROC]) was 0.89 [29]. Use of the urine cystatin C level for early detection of AKI after cardiac surgery allows for the diagnosis of tubular damage and dysfunction [30]. Serum cystatin C is a useful biomarker for AKI in patients in the intensive care unit (ICU) [31,32] with contrast-induced AKI [29,33]. In one study, preoperative serum cystatin C levels were significantly higher in children with UPJO compared to controls and decreased after surgery (Table 1) [26], and the AUC-ROC value of serum cystatin C indicating UPJO was 0.72 (Table 1) [26,34]. In another study, serum cystatin C levels increased in adults with ureteral calculi as hydronephrosis increased and differed significantly between patients with no and mild hydronephrosis, while SCr levels did not [35]. Multivariate logistic regression showed that only the serum cystatin C level was an independent risk factor for hydronephrosis. By contrast, the urine cystatin C level is less useful as a UUTO biomarker. In two independent studies of children with UUTO, urine cystatin C levels did not differ between patients and controls (Table 1) [24,36].

### 5.2. Biomarkers of Renal Tubular Damage

#### 5.2.1. NGAL

Human NGAL, a ubiquitous 25 kDa protein, was initially isolated from human neutrophils [37]. NGAL is expressed in small amounts in cells other than neutrophils, including lung, spleen, and kidney cells, and is thought to inhibit bacterial growth, scavenge iron, and induce epithelial growth [38]. NGAL can be secreted by epithelial cells, and it is markedly elevated in patients exhibiting an inflammatory immune response and defects in lipid metabolism, intracellular iron transport, renal tubular repair, or differentiation of kidney progenitor cells into tubular epithelial cells [39]. In the kidney, NGAL is secreted into the urine from the ascending limb of the loop of Henle to the collecting ducts, being synthesized in the distal nephron [40]. NGAL is small, freely filtered, and easily assayed in urine. The urine NGAL level is an early and sensitive biomarker of kidney injury [41]. The serum or urine NGAL level is a clinically useful biomarker of various types of AKI, including AKI after kidney transplantation [42], contrast medium-induced AKI [43], and AKI in critical care settings [44]. In children with UUTO, urine NGAL levels are significantly higher in bladder urine and/or renal pelvic urine compared to controls, correlate inversely with worsening obstruction, and decrease after surgery (Table 1) [26,36,45,46,47,48,49]. The AUC-ROC value for UUTO in children is 0.61–0.90 for bladder urine NGAL [26,36,45,46,47,48]. In adults with UUTO, the urine NGAL level increases in those with obstructive nephropathy (AUC-ROCs of 0.70 for bladder urine and 0.76 for renal pelvic urine) and decreases after relief of the obstruction [23,50,51]. Serum NGAL levels are significantly higher than in controls [45,50]. However, the use of NGAL as a biomarker of kidney injury induced by UUTO has several limitations. Age affects the predictive performance: NGAL better predicts AKI in children than in adults [52]. Serum and urine NGAL levels may be influenced by conditions other than UUTO, including chronic hypertension, systemic infection, inflammation, anemia, hypoxia, or malignancy [53,54,55]. The many sources of NGAL can render it difficult to identify the underlying pathology [40].

#### 5.2.2. MCP-1

MCP-1, a 13 kDa protein, is a potent attractant of monocytes and a member of the CC subfamily [56]. It is produced by many types of cells, including epithelial, endothelial, and smooth muscle cells; fibroblasts, astrocytes, and monocytes; and microglial cells. MCP-1 recruits monocytes, memory T-cells, and dendritic cells to sites of tissue injury and infection; monocytes and macrophages are the major sources of MCP-1 [57]. MCP-1 mRNA is undetectable in the normal kidney, but MCP-1 gene expression is markedly increased at the tubulointerstitial level in UPJO biopsy samples and correlates with the extent of monocyte infiltration [58,59]. In one experimental study, mice deficient in MCP-1 exhibited significantly decreased survival and increased renal damage after ischemia/reperfusion-induced renal tubular injury in the absence of macrophage accumulation [60]. Kidneys and primary tubular epithelial cells from such mice exhibited increased apoptosis after ischemia, which indicates that MCP-1 protects the kidney from the acute inflammatory response that develops after kidney injury. MCP-1 is one of the most promising biomarkers of kidney injury [61]. Elevated MCP-1 levels are associated with immune system-mediated kidney injury [62,63], diabetic nephropathy [64], and autosomal-dominant polycystic kidney disease [65]. In a mouse model of UUTO, serum and urine MCP-1 levels increased significantly compared to those of control mice [66]. mRNA expression and urinary excretion of MCP-1 correlate with the extent of the obstruction, subsequent renal damage, and hydronephrosis. Urine levels of MCP-1 decrease after release of the obstruction [67]. Urine MCP-1 levels are significantly increased in UPJO groups compared to controls and fall significantly after surgery [24,36,46,59,68,69,70]. The AUC-ROC values in terms of the presence of UUTO are 0.70–0.93 for bladder urine MCP-1 and 0.89 for renal pelvic urine MCP-1 (Table 1) [24,36,46,68,69]. An inverse correlation is evident between the level of MCP-1 in renal pelvic urine and mertiatide clearance by the affected kidney [59,69]. Urine MCP-1 levels usefully distinguish between UPJO (which requires pyeloplasty) and the absence of an obstructive dilation of the renal pelvis. They can be used to monitor the resolution of kidney damage after surgery for UUTO [36]. Serum MCP-1 levels increase in patients with AKI after cardiac surgery and in those with chronic kidney damage [60,71,72], but no study has yet evaluated the serum MCP-1 level as a biomarker of UUTO in a clinical setting.

#### 5.2.3. KIM-1

KIM-1 is a type I membrane protein of 104 kDa composed of a 14 kDa membrane-bound fragment and a 90 kDa soluble portion [73]. It was isolated from T-cells, exhibits various functions, and was termed T-cell immunoglobulin-and-mucin-domain-containing molecule-1 (TIM-1) [73]. Normal kidney tissue rarely expresses KIM-1, but kidneys acutely injured by ischemia, hypoxia, toxicity, or renal tubular interstitial/polycystic kidney disease do [74]. The ectodomain of KIM-1 (90 kD) is cleaved by matrix metalloproteinases and is found in urine after injury to the kidney proximal tubules [75]. Acute KIM-1 overexpression in proximal, renal tubular epithelial cells after ischemia, hypoxia, or toxicity promotes the transformation of these cells into semi-professional phagocytic cells. KIM-1 is a phosphatidylserine receptor of the liposome surface and identifies both apoptotic bodies and phosphatidylserine, triggering further phagocytosis [74,76]. The upregulation of KIM-1 by injured tubular epithelial cells facilitates the clearance of apoptotic cells, protecting against AKI. Apart from mediating phagocytosis, KIM-1 assists in repairing injury to cells [77]. It is a valuable biomarker of AKI. Urine and/or serum KIM-1 levels increase after ischemic kidney injury [75] and in patients with diabetic nephropathy [78], IgA nephropathy [79], and kidney injury after renal transplantation [80]. In an animal model, serum and urine KIM-1 levels were useful for the early diagnosis of obstructive nephropathy-induced AKI [81,82]. In a mouse model, serum KIM-1 levels increased after UUTO, peaking on day 3, and remained detectable for 14 days [82]. In a rat model, the urine KIM-1 level began to increase on day 1 after UUTO and remained high until day 7 [81]. In children with UUTO, urine KIM1 levels correlated inversely with worsening obstruction and decreased after surgery [34,36,45,48,49]. The AUC-ROC value for the prediction of childhood UUTO is 0.65–0.89 for the bladder urine KIM-1 level (Table 1) [34,36,45,49]. In adults with UUTO, the urine KIM-1 level is a useful marker of obstructive nephropathy (AUCs of 0.57–0.73 for bladder urine and 0.88 for renal pelvic urine) [23,50,51]. Xie found that the urine KIM-1 level after surgery to treat UUTO predicted renal function deterioration [83].

#### 5.2.4. NAG

NAG, a 130–140 kDa protein, is a lysosomal enzyme distributed in various human tissue [84]. NAG is not filtered through the glomeruli. In the kidney, it is found predominantly in lysosomes of proximal tubular cells. The small amount of NAG normally present in the urine is exocytosed by these cells. Although the function of NAG in the kidney remains unknown, it is a marker of tubular cell function or damage [85]. Increased NAG excretion in urine is caused exclusively by proximal tubular cell injury. Accumulating evidence indicates that urine NAG levels correlate with exposure to nephrotoxic drugs, delayed allograft nephropathy, diabetic nephropathy, and AKI [85]. Urine NAG levels are elevated in patients with upper urinary tract infection, nephrolithiasis, and reflux nephropathy [86,87]. In children with UUTO, urine NAG levels were significantly higher in those with hydronephrosis (with or without a vesicoureteral reflux) than healthy controls or cystitis patients [88,89]. The NAG level in renal pelvic urine is 7-fold higher and that in bladder urine 1.7-fold higher than in normal controls [89]. Mohammad found that the AUC-ROC value for bladder NAG was 0.67 in children with UUTO [68]. Skalova reported that although the urine NAG level was significantly higher in patients with hydronephrosis compared to healthy controls, there were no differences between children with unilateral and bilateral hydronephrosis and no correlation between the urine NAG level and the grade of hydronephrosis [90]. In summary, urine NAG levels usefully detect childhood UUTO but do not reflect its severity. In one study of adults with UUTO, levels of NAG in bladder and renal pelvic urine were 2.5- and five-fold higher than those of normal controls (AUC-ROC values of 0.74 for bladder urine and 0.91 for renal pelvic urine) and decreased after treatment [23].

#### 5.2.5. L-FABP

L-FABP, which is expressed by both the normal and diseased human kidney, has been found in both the convoluted and straight portions of human proximal tubules [91]. Mammalian intracellular FABP is a 14 kDa protein encoded by a member of a large multigene family within a superfamily of lipid-binding proteins [92]. Nine tissue-specific FABPs have been identified: L (liver), I (intestinal), H (muscle and heart), A (adipocyte), E (epidermal), IL (ileal), B (brain), M (myelin), and T (testis). All FABPs primarily regulate fatty acid metabolism and intracellular transport [93]. L-FABP is expressed not only in the liver but also in the intestine, pancreas, stomach, lung, and kidney [94]. Serum and/or urine L-FABP levels are useful biomarkers of kidney injury after renal transplantation [95], in critical care patients with AKI [96], and in those with contrast-induced AKI [97] and diabetic nephropathy [98]. However, the utility of L-FABP for predicting UUTO remains controversial. Xie found that urine L-FABP levels after UUTO surgery predicted the deterioration of renal function [83]. Furthermore, in one study of patients with vesicoureteral refluxes, the urine L-FABP level was significantly higher than in controls [99]. However, Noyan found that urine L-FABP levels did not differ significantly between children with hydronephrosis and controls [48].

### 5.3. Novel Biomarkers of UUTO

#### 5.3.1. Vanin-1

Vanin-1, a 53 kDa protein, is expressed in the brush borders of the proximal tubule of the kidney [100]. By catabolizing pantetheine to cysteamine and pantothenic acid (a precursor of coenzymes), it has roles in metabolism and energy production. The function of kidney vanin-1 remains to be established. However, the fact that vanin-1 is located specifically in the brush borders suggests that the enzyme plays a pivotal role in pantothenic acid salvage and recycling. The proximal tubular cells bear microvilli with large apical surface areas within which many transporters and channels are found [101]. Vanin-1 in cellular membranes is anchored to glycosylphosphatidylinositol. The anchor may be cleaved and soluble vanin-1 then secreted or released into the extracellular matrix in response to various stimuli [102].

Urine vanin-1 levels are increased in patients with drug-induced AKI [103] and UUTO [23], and in rat models with high salt-induced kidney damage [104], diabetic nephropathy [105], and UUTO [102]. UUTO inhibits urine flow and increases intratubular pressure, causing renal tubular damage. Vanin-1 is then secreted into the urine by renal tubular cells. The level of vanin-1 in renal pelvic urine correlates highly with the severity of urinary tract obstruction [23]. The level of vanin-1 in renal pelvic urine is highly predictive (AUC-ROC value 0.98) of adult UUTO, more predictive than NGAL, KIM-1, or NAG levels. Vanin-1 levels decrease following UUTO relief in patients with moderate to severe UUTO [23].

#### 5.3.2. α-Glutathione S-Transferase (GST)

GST, a 28 kDa protein, is a cytosolic enzyme. The isoforms α and π (α-GST, π-GST) are typical of the human kidney [106]. α-GST is expressed in proximal tubular epithelial cells, and π-GST is expressed in distal tubular epithelial cells [45]. Both isoforms of GST are released from injured cells into the urine and were recently suggested to be promising biomarkers of kidney injury [107] in the context of cyclosporine-induced nephrotoxicity, cadmium exposure, administration of nephrotoxic antibiotics, acute transplant rejection [106], and critical illness in the ICU [106,108]. Recently, the utility of the α-GST and π-GST level in terms of predicting UUTO was explored. Children with UPJO exhibited significantly higher urinary α-GST excretion than controls. Urinary AUC-ROC values for UUTO detection were 0.90 for α-GST and 0.3 for π-GST. The predictive performance of α-GST was superior to that of urinary NGAL or KIM-1 [45].

#### 5.3.3. Tissue inhibitor of metalloproteinases-2 (TIMP-2)/ insulin-like growth factor-binding protein 7 (IGFBP7)

TIMP-2 and IGFBP7 are new AKI biomarkers. In 2014, Food and Drug Administration approved TIMP-2/ IGFBP7 to be used in ICU patients to predict the risk of developing moderate to severe AKI [109]. TIMP-2 has a molecular weight of approximately 24 kDa and IGFBP7 has a molecular mass of 29 kDa [110]. Both of them are expressed and secreted by renal tubular cells, and involved in G1 cell cycle arrest during the early phases of cellular stress or injury caused by various insults (e.g., sepsis, ischemia, oxidative stress, and toxins) [111]. TIMP-2/ IGFBP7 shows the best accuracy among AKI biomarkers in patients with various types of AKI condition including AKI after kidney transplantation and AKI in critical care settings, sepsis and platinum-based chemotherapy, and chronic kidney damage induced by diabetes mellitus and congestive heart failure [112,113]. However, no study has yet evaluated the TIMP-2/ IGFBP7 as a biomarker of UUTO in an animal study or a clinical setting, and therefore it is urgently necessary.

### 5.4. Comparison of Biomarkers

Several studies have compared the utility of NAGL, KIM-1, and/or L-FABP levels as biomarkers of childhood UUTO. In studies, urine and/or serum NAGL levels outperformed urine KIM-1 or L-FABP levels [48,49]. In one study, striking increases in serum and urine NGAL levels were evident in patients with obstructive nephropathy, whereas urine KIM-1 levels did not differ significantly between patients and controls, which suggests that KIM-1 is not sensitive in this setting [50]. In another study, urine NGAL levels were significantly higher in patients with both hydronephrosis and obstruction than in those with hydronephrosis but no obstruction or normal controls. Urine KIM-1 and L-FABP levels did not differ significantly among the groups [48]. Patients with renal colic who also exhibited hydronephrosis had significantly higher urine NAG and NGAL, but not KIM-1, levels than did patients without hydronephrosis [114]. In a mouse model of ischemia/reperfusion kidney injury, serum and urine KIM-1 levels increased during the acute phase and declined gradually in the chronic phase, while serum and urine NGAL levels increased continuously during the transition from AKI to chronic kidney disease, which suggests that NGAL is a valuable biomarker in this setting [41]. This may explain why NGAL is a better biomarker of UUTO than KIM-1. However, one predictive model of worsening kidney function after surgery found that urine KIM-1 and L-FABP levels more reliably predicted kidney deterioration after surgical removal of ureteral stones than did urine NGAL levels [83].

MCP-1 is one of the best biomarkers of childhood UUTO. In one study, urine MCP-1 levels were significantly higher in a pyeloplasty group than a non-obstruction group, while urine NGAL and KIM-1 levels did not differ significantly between the groups [36]. In another study, urine MCP-1 levels were significantly higher in patients with hydronephrosis who required surgery than in those who did not; urine NAG levels did not differ significantly between the groups [68]. In one study, the AUC-ROC values of bladder urine MCP-1 and NGAL in children with UPJO were 0.89 and 0.90, respectively, higher than those of bladder urine interleukin-6 (0.78) or TGF-β1 (0.67) [46].

### 5.5. Panel Assessment of Biomarkers

No single biomarker is specific for UUTO, and given the multifactorial nature of obstruction, not all obstructions can be identified using a single biomarker [24]. In children with UPJO, combined NGAL/MCP-1 assessment improved diagnostic performance compared to assessment of either biomarker alone [46]. In another study on such children, the AUC-ROC values were 0.63 for SCr, 0.72 for serum cystatin C, 0.80 for urinary NGAL, 0.70 for urinary KIM-1, and 0.70 for urinary cystatin C. The AUC-ROCs of combinations of these biomarkers were higher than those of the single biomarkers, being highest (0.88) for urinary NGAL + urinary cystatin C + serum cystatin C [34]. In critically ill patients with AKI, a combination of urine NGAL and L-FABP levels, sepsis status, blood lactate level, and stratification using the Acute Physiology and Chronic Health Evaluation score improved AKI predictive performance (AUC-ROC 0.94) compared to NGAL alone (AUC-ROC 0.86) or L-FABP alone (AUC-ROC 0.84) [43]. In a model predicting worsening kidney function after surgery in UUTO patients, the AUC-ROC of the preoperative combination of urinary biomarkers L-FABP, KIM-1, and NGAL was 0.97, higher than the highest AUC of a single biomarker (0.91 for L-FABP) [83].

## 6. Current Limitations and Future Directions

Most obstructions of the upper urinary tract are unilateral. Reduced glomerular filtration in the affected kidney and the obstruction per se decreases the amount of any biomarker that reaches the bladder, which explains why the biomarker AUC-ROCs of bladder urine are generally lower than those of renal pelvic urine [23,24]. Combinations of serum and bladder biomarker levels may thus be optimal. However, serum values of biomarkers have been less studied than urine values in UUTO patients. Only serum cystatin C and NGAL levels are clinically useful. Serum markers that are highly predictive of UUTO should be sought. Combinations of serum and urinary biomarkers facilitate the diagnosis of UUTO, risk stratification, clinical decision making, and monitoring.

Age affects the predictive performance of biomarkers [52]. Acute and/or chronic kidney injury is a condition frequently found in elderly population with comorbidities, which may alter the value of biomarkers. This would be the reason why NGAL better predicts AKI in children than in adults [52].

Both urinary and serum UUTO biomarkers lack specificity. Increases may be associated with conditions other than UUTO or even non-kidney conditions. Increases in MCP-1 are associated with liver cirrhosis [115] and sleep apnea syndrome [116], increases in NGAL are associated with cardiovascular ischemia, heart failure, atherosclerosis, and pneumonia [117,118], and increases in L-FABP are associated with various liver diseases [119,120]. However, panel assessment may compensate for the lack of specificity.

## 7. Conclusions

Renal scans are standard in evaluations of the presence and severity of UUTO, but they are expensive, are not always available, and expose patients to radiation. Many urinary and serum biomarkers have been studied in children and adults with UUTO. MCP-1 and NGAL, the most extensively studied, are the most likely to be optimal. Recently, novel biomarkers (vanin-1 and α-GST) have outperformed traditional biomarkers in terms of evaluating UUTO, but further work must explore whether this is the case in all UUTO settings. No single biomarker is adequately sensitive or specific. Panel assessment affords mutual biomarker compensation and improves predictive performance. The obstruction per se and reduced glomerular filtration in the affected kidney decrease the amount of any biomarker reaching the bladder, limiting the performance of bladder urine biomarkers. However, combinations of serum and bladder urinary biomarkers improve performance. Panel assessment of urinary and serum biomarkers facilitates the diagnosis of UUTO, risk stratification, clinical decision making, and monitoring.

## Figures and Tables

**Figure 1 ijms-21-05490-f001:**
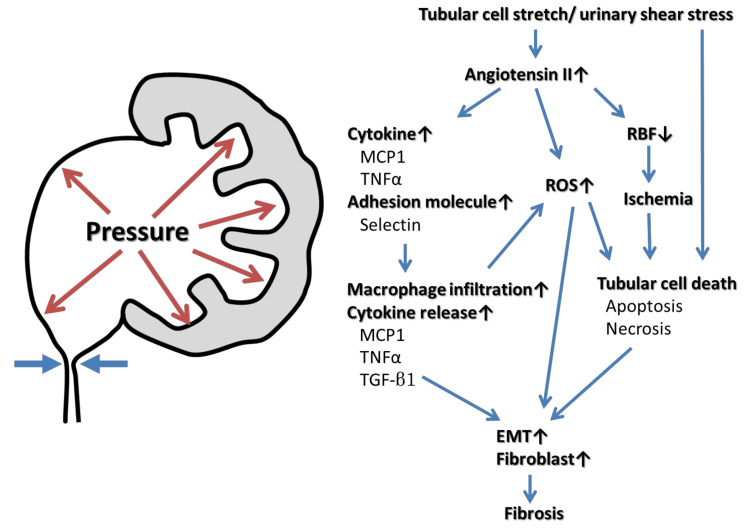
Mechanisms of UUTO causing kidney injury and fibrosis.

**Table 1 ijms-21-05490-t001:** Urinary and serum biomarkers for pediatric and adults UUTO.

Biomarkers		Author, Ref.	Publish Year	Disease	Pts Number	Mean or Median Age of Pts	Laterality of Affected Kidneys	Source of Samples	Comparison of Values, p-Value	AUC- ROC	Comparison of Group
Urinary NGAL	Ped	Pavlaki A, 26	2020	UPJO	22 Obst, 19 Non-obst, 17Cts	3.0 months	Uni	Bl	0.01, Obst vs. Cts	0.61	Obst + Non-obst vs. Cts
		Kostic D, 34	2019	HN	37 Obst, 45 Cts	5.0 months	Uni + Bi	Bl	NA	0.80	Obst vs. Cts
		Yu L, 46	2019	UPJO	17 Pts, 17 Cts	NA	Uni + Bi	Bl	0.0004, Pts vs. Cts	0.90	Pts vs. Cts
		Bienias B, 45	2018	UPJO	28 Obst, 17 Non-obst, 21 Cts	11 years	Uni	Bl	<0.05, Obst vs. Cts	0.66	Obst vs. Cts
		Gupta S, 47	2018	UPJO	30 Pts, 15 Cts	4.7 years	Uni	Bl	0.0009, Pts vs. Cts	0.80	Pts vs. Cts
		Karakus S, 36	2016	UPJO	13 Obst, 14 Non-obst, 9 Cts	3.9 years	Uni	Bl	0.032, Obst vs. Cts	0.85	Obst vs. Cts
		Noyan A, 48	2015	UPJO	26 Pts, 36 Non-obst, 20 Cts	21 months	NA	Bl	<0.05, Obst vs. Cts	0.68	Obst vs. Cts
		Madsen MG, 24	2013	UPJO	24 Pts, 13 Cts	6.5 years	Uni	Bl	NS, Pts vs. Cts	NA	NA
								Rp	<0.05, Pts vs. Cts	NA	NA
		Wasilewska A, 49	2011	UPJO	20 Obst, 20 Non-obst, 25 Cts	2.2 years	Uni	Bl	<0.01, Obst vs. Cts	0.81	Obst vs. Cts
								Rp	<0.01, Obst vs. Cts	NA	NA
	Adul	Washino S, 23	2019	UUTO	28 Pts, 21 Cts	54 years	Uni	Bl	<0.05, Pts vs. Cts	0.70	Pts vs. Cts
								Rp	<0.01, Pts vs. Cts	0.76	Pts vs. Cts
		Olvera-Posada D, 51	2017	HN	24 Obst 20 Non-obst, 11Cts	58.5 years	Uni	Bl	0.009, Obst vs. Cts	NA	NA
		Urbschat A, 50	2014	Ureteral calculi	53 Pts, 52 Cts	44 years	Uni	Bl	<0.05, Pts vs. Cts	NA	NA
Urinary MCP-1	Ped	Yu L, 46	2019	UPJO	17 Pts, 17 Cts	NA	Uni + Bi	Bl	0.0005, Pts vs. Cts	0.89	Pts vs. Cts
		Karakus S, 36	2016	UPJO	13 Obst, 14 Non-obst, 9 Cts	3.9 years	Uni	Bl	0.002, Obst vs. Cts	0.93	Obst + Non-obst vs. Ct
		Mohanmmadjafari H, 68	2014	HN	24 Obst, 18 Non-obst	6.5 years	Uni + Bi	Bl	0.012, Obst vs. Cts	0.73	Obst vs. Non obst
		Madsen MG, 24	2013	UPJO	28 Pts, 13 Cts	6.5 Years	Uni	Bl	<0.05, Pts vs. Cts	0.78	Pts vs. Cts
								Rp	<0.05, Pts vs. Cts	0.89	Pts vs. Cts
		Taranta-Janusz K, 69	2012	HN	15 Obst, 21 Non-obst, 19 Cts	0.25 years	Uni	Bl	<0.05, Obst vs. Cts	0.70	Pts vs. Cts
								Rp	<0.01, Obst vs. Cts	NA	NA
		Bartoli F, 70	2011	UPJO	12 Obst, 36 Non-obst, 30 Cts	NA	NA	Bl	<0.001, Obst vs. Cts	NA	NA
		Grandaliano G, 59	2000	UPJO	24 Pts, 15 Cts	NA	NA	Bl	<0.01, Pts vs. Cts	NA	NA
Urinary KIM-1	Ped	Kostic D, 34	2019	HN	37 Obst vs. 45 Cts	5.0 months	Uni + Bi	Bl	NA	0.70	Obst vs. Cts
		Bienias B, 45	2018	UPJO	28 Obst, 17 Non-obst, 21 Cts	11 years	Uni	Bl	<0.05, Obst vs. Cts	0.65	Obst vs. Cts
		Karakus S, 36	2016	UPJO	13 Obstr, 14 Non-obst, 9 Cts	3.9 years	Uni	Bl	0.001, Obst vs. Cts	0.89	Obst + Non-obst vs. Ct
		Noyan A, 48	2015	UPJO	26 Pts, 36 Non-obst, 20 Cts	21 months	NA	Bl	NS, Obst vs. Cts	NA	Obst vs. Cts
		Wasilewska A, 49	2011	UPJO	20 Obst, 20 Non-obst, 25 Cts	2.2 years	Uni	Bl	<0.01, Obst vs. Cts	0.80	Obst + Non-obst vs. Ct
								Rp	<0.01, Obst vs. Cts	NA	NA
	Adul	Washino S, 23	2019	HN	28 Pts, 21 Cts	54 years	Uni	Bl	NS, Pts vs. Cts	0.57	NA
								Rp	<0.01, Pts vs. Cts	0.88	NA
		Olvera-Posada D, 51	2017	HN	24 Obst 20 Non-obst, 11Cts	58.5 years	Uni	Bl	0.02, Obst vs. Cts	0.73	Obst vs. Cts
		Urbschat A, 50	2014	Ureteral calculi	53 Pts, 52 Cts	44 years	Uni	Bl	NS, Pts vs. Cts	NA	NA
Urinary NAG	Ped	Skalova S, 84	2007	HN	31 Pts, 262 reference Cts	2.3 years	Uni + Bi	Bl	0.002, Pts vs. Cts	NA	NA
		Mohanmmadjafari H, 68	2014	HN	24 Obst, 18 Non-obst	6.5 years	Uni + Bi	Bl	NS, Obst vs. Non-obst	0.67	Obst vs. Non-obst
	Adul	Washino S, 23	2019	HN	28 Pts, 21 Cts	54 years	Uni	Bl	<0.01, Pts vs. Cts	0.74	Pts vs. Cts
								Rp	<0.001 Pts vs. Cts	0.91	Pts vs. Cts
Urinary L-FABP	Ped	Noyan A, 48	2015	UPJO	26 Pts, 36 Non-obst, 20 Cts	21 months	NA	Bl	NS, Obst vs. Cts	NA	NA
Urinary Vanin-1	Adul	Washino S, 23	2019	HN	28 Pts, 21 Cts	54 years	Uni	Bl	<0.05, Pts vs. Cts	0.63	Pts vs. Cts
								Rp	<0.0001, Pts vs. Cts	0.98	Pts vs. Cts
Urinary α-GST	Ped	Bienias B, 45	2018	UPJO	28 Obst, 17 Non-obst, 21 Cts	11 years	Uni	Bl	<0.05, Obst vs. Cts	0.90	Obst vs. Cts
Urinary CyC	Ped	Kostic D, 34	2019	HN	37 Obst vs. 45 Cts	5.0 months	Uni + Bi	Bl	NA	0.71	Obst vs. Cts
		Karakus S, 36	2016	UPJO	13 Obst, 14 Non-obst, 9 Cts	3.9 years	Uni	Bl	NS, Obst vs. Cts	NA	NA
		Madsen MG, 24	2012	UPJO	24 Pts, 13 Cts	8.0 years	Uni	Bl	NS, Pts vs. Cts	NA	NA
								Rp	NS, Pts vs. Cts	NA	NA
Serum NGAL	Ped	Bienias B, 45	2018	UPJO	28 Obst, 17 Non-obst, 21 Cts	11 years	Uni	S	<0.05, Obst vs. Cts	1.00	Obst vs. Cts
	Adul	Urbschat A, 50	2014	Ureteral calculi	53 Pts, 52 Cts	44 years	Uni	S	<0.01, Pts vs. Cts	NA	NA
Serum CyC	Ped	Pavlaki A, 26	2020	UPJO	22 Obst, 19 Non-obst, 17Cts	3 months	Uni	S	0.01, Obst vs. Cts	0.72	Obst + Non-obst vs. Cts
		Kostic D, 34	2019	HN	37 Obst vs. 45 Cts	5.0 months	Uni + Bi	S	NA	0.72	Obst vs. Cts
	Adul	Mao W, 35	2020	Ureteral calculi	160 HN, 40 Non-HN	52 years	Uni	S	<0.001, HN vs. Non-HN	0.66	HN vs. Non-HN

Abbreviation: NGAL: neutrophil gelatinase-associated lipocalin; MCP-1: monocyte chemotactic protein-1; KIM-1: kidney injury molecule 1; NAG: N-acetyl-b-D-glucosaminidase; L-FABP: liver type fatty acid-binding protein; α-GST: α-glutathione S-transferases; CyC: Cystatin C, Ped: pediatrics; Adul: adults; UPJO: ureteropelvic junction obstruction; HN: hydronpehrosis; Pts: patients; Obst: patients with obstructive hydronephrosis; Non-obst: patients with non-obstructive hydronephrosis; Cts: controls; NA: not assessed; Uni: unilateral; Bi: bilateral; Bl: bladder; Rp: renal pelvis; S: serum; NS: not significant; AUC-ROC: area under curve in receiver operating characteristics.

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
