# Peer review of "Roles Played by Biomarkers of Kidney Injury in Patients with Upper Urinary Tract Obstruction"

_ijms, 2020, doi:10.3390/ijms21155490_

Round 1
Reviewer 1 Report
Thank you for giving me the possibility to evaluate the paper entitled “Roles played by biomarkers of kidney injury in patients with upper urinary tract obstruction”. In my opinion the manuscript does not fit the aim of International Journal of Molecular Sciences. The journal “provides an advanced forum for molecular studies in biology and chemistry, with a strong emphasis on molecular biology and molecular medicine”. Moreover, the scope of the journal is dealing with “i) fundamental theoretical problems of broad interest in biology, chemistry and medicine; ii) breakthrough experimental technical progress of broad interest in biology, chemistry and medicine; iii) application of the theories and novel technologies to specific experimental studies and calculations. The authors wrote a narrative review about relationship between clinical features and biomarkers in obstructive acute kidney injury. Finally, I suggest to the authors to carefully check abbreviation in the text and to discuss the fact that acute kidney injury is a conditions frequently found in elderly population with comorbidities, and the latter could alter the value of biomarkers.
Author Response
To reviewer 1
I would like to thank you very much to review this manuscript. As the reviewer pointed, this review is mainly composed of clinical studies, but this also includes molecular pathways on how kidney injury occurs in UUTO and how biomarker proteins are induced and function in urinary obstruction or kidney injury. This also includes the results of animal studies when clinical data are not sufficient. In my opinion, this review is composed of a combination of clinical and basic studies rather than just clinical studies or basic molecular studies. I would say this review can help the understanding of kidney biomarkers of UUTO for both clinicians and basic researchers and connect these.
The parts we changed in this manuscript are yellow-highlighted.
Point by point response
- Abbreviations: I checked abbreviations in this text and revised accordingly (yellow highlighted).
- Acute kidney injury is a condition frequently found in the elderly population with comorbidities, which may alter the value of biomarkers: I added some comments about this in the Limitation section. Please find out it on page 18, the first paragraph.
Reviewer 2 Report
This is an excellent and timely review by Satoshi Washino and colleagues on biomarkers for urinary tract obstruction associated kidney injury.
Specific comments:
The authors very well summarize the current literature on this topic. However, they have not mentioned or discussed TIMP2/IGFBP7 anywhere in the manuscript. A new AKI biomarker “NephroCheck,” has received Food and Drug Administration approval for detection of kidney injury based on a large number of clinical trials. The test is a composite of two independent biomarker proteins: the tissue metalloproteinase inhibitor 2 (TIMP2) and the IGF-binding protein 7 (IGFBP7). Because TIMP2 and IGFBP7 can contribute to AKI-induced cell cycle inhibition, NephroCheck has also been described as a kidney “cell cycle arrest” biomarker. Appropriate discussion of the utility of TIMP2/IGFBP7 for urinary tract obstruction associated kidney injury would be critical.
Author Response
I would like to thank you very much to review this manuscript. As the reviewer pointed, TIMP-2/IGFBP7 is the AKI biomarker to be approved by the FDA. I added some comments about TIMP-2/IGFBP7 to this manuscript. However, TIMP-2/IGFBP7 has not been studied in the field of UUTO. Definitely we need to assess the usefulness of TIMP-2/IGFBP7 in UUTO in the future. I added a paragraph about TIMP-2/IGFBP7 to our manuscript. Please find out it on page 13, the first paragraph.
Round 2
Reviewer 1 Report
The revised version of the paper is very similar to the previous one. My opinion has not changed, the article does not fit the aim of the journal.